# A Holistic Approach to Producing Anti-*Vibrio* Metabolites by an Endosymbiotic Dinoflagellate Using Wastewater from Shrimp Rearing

**DOI:** 10.3390/microorganisms12081598

**Published:** 2024-08-06

**Authors:** Carlos Yure B. Oliveira, Jéssika L. Abreu, Barbara C. Brandão, Deyvid Willame S. Oliveira, Pedro Rodrigues de Sena, Weverson Ailton da Silva, Evando S. Araújo, Leonardo R. Rörig, Gisely Karla de Almeida Costa, Suzianny Maria B. C. Silva, Marius N. Müller, Giustino Tribuzi, Alfredo O. Gálvez

**Affiliations:** 1Laboratory of Live Food Production, Department of Fisheries and Aquaculture, Federal Rural University of Pernambuco, Recife 52171-900, PE, Brazil; jessik.labreu@gmail.com (J.L.A.); soaresbarbara07@hotmail.com (B.C.B.); deyvidwillame@gmail.com (D.W.S.O.); pedro.rsenar@gmail.com (P.R.d.S.); alfredo_oliv@yahoo.com (A.O.G.); 2Laboratory of Phycology, Department of Botany, Federal University of Santa Catarina, Florianópolis 88040-900, SC, Brazil; leororig@gmail.com; 3Fishery Resources and Engineering Postgraduate Program, State University of West Paraná, Toledo 85903-000, PR, Brazil; pescailton@gmail.com; 4Research Group on Electrospinning and Nanotechnology Applications, Department of Materials Science, Federal University of San Francisco Valley, Juazeiro 48902-300, BA, Brazil; evando.araujo@univasf.edu.br; 5Laboratory of Aquatic Animal Health, Department of Fisheries and Aquaculture, Federal Rural University of Pernambuco, Recife 52171-900, PE, Brazil; gisely.costa@ufrpe.br (G.K.d.A.C.); suzianny.silva@ufrpe.br (S.M.B.C.S.); 6Department of Oceanography, Federal University of Pernambuco, Recife 50740-600, PE, Brazil; marius.muller@ufpe.br; 7Department of Food Science and Technology, Federal University of Santa Catarina, Florianopólis 88034-801, SC, Brazil; giustino.tribuzi@ufsc.br

**Keywords:** aquaculture, dinoflagellate, peridinin, symbiodiniaceae

## Abstract

The aquaculture industry requires green solutions to solve several environmental challenges, including adequate wastewater remediation and natural drug applications to treat bacteria- and virus-related diseases. This study investigated the feasibility of cultivating the dinoflagellate *Durusdinium glynnii* in aquaculture wastewater from shrimp rearing in a synbiotic system (AWW-SS), with different dilutions of f/2 medium (FM). Interestingly, *D. glynnii* demonstrated enhanced growth in all AWW–SS treatments compared to the control (FM). The highest growth rates were achieved at AWW-SS:FM dilutions of 75:25 and 50:50. The removal of total nitrogen and total phosphorus reached 50.1 and 71.7%, respectively, of the crude AWW–SS. Biomass extracts of *D. glynnii* grown with AWW–SS were able to inhibit the growth of the bacteria *Vibrio parahaemolyticus* (inhibition zone of 10.0 ± 1.7 mm) and *V. vulnificus* (inhibition zone of 11.7 ± 1.5 mm). The presented results demonstrate that the dinoflagellate *D. glynnii* is a potential candidate for the development of circularity for sustainable aquaculture production, particularly by producing anti–*Vibrio* compounds at a near-zero cost.

## 1. Introduction

Aquaculture is a key component for achieving the Sustainable Development Goals (SDGs) set out by the United Nations in the 2030 Agenda, particularly for the SDGs 1—no poverty, 2—zero hunger, 3—good health and well–being, 8—decent work and economic growth, 12—responsible consumption and production, 13—climate action, and 14—life below water [1,2]. In recent years, aquaculture fish production has surpassed traditional fisheries, and sustainable aquaculture can contribute to restoring overexploited fish stocks around the world without compromising food security [3]. However, new practices and methods need to be developed and/or improved to solve the longstanding problems of aquaculture, such as the use of alternative protein and oil sources suitable for fish nutrition [4,5], appropriate treatment of wastewaters rich in nitrogen and phosphorus [6,7], and the adoption of sustainable drugs to treat bacterial and virus related diseases [8]. Regarding fish diseases, vibriosis is one of the major threats to marine aquaculture farmers worldwide, resulting in high fish mortalities and economic losses [9]. The genus *Vibrio* is composed of Gram-negative pathogenic bacteria that are widely distributed in marine environments. Furthermore, some *Vibrio* species have demonstrated resistance to traditional antibiotics applied in aquaculture, such as cotrimoxazole, chloramphenicol, and streptomycin. Therefore, new anti–*Vibrio* agents with a low resistance risk are of great interest to prevent *Vibrio*-related economic losses in the aquaculture sector [9,10].

Microalgae are considered a promising alternative to solve these aquaculture issues [11]. This is because they can be used to efficiently treat wastewater, absorbing up to 99% of nitrogen and phosphorus sources [12,13], and to produce valuable biomass that can be rich in lipids and proteins, as well as other bioactive metabolites, suitable for fish nutrition [14]. This circular aquaculture approach enables the recycling of waste in a holistic setting with zero or near-zero residue production. Circular bioeconomy is based on the application of green and sustainable chemistry principles to replace fossil-based materials with biologically-based and renewable resources. The circular approach in aquaculture has the potential to improve profitability and sustainability through the valorization of by-products and wastes (solids and liquids) [15]. Although several studies have already proven the ability of microalgae to grow in different types of aquaculture wastewater (e.g., [16,17,18]), few studies have investigated the application of this biomass as an aquaculture input to demonstrate the effectiveness of a circular and holistic setting.

Microalgae are capable of synthesizing primary and secondary metabolites and, according to Lopes da Silva et al. [19], contribute to the global bioeconomy due to their ability to produce marketable value-added products using liquid, gaseous, and solid wastes as resource. Primary metabolites are those necessary for cell growth, such as protein, carbohydrates, and lipids, while secondary metabolites have vital functions in ecological interactions and adaptive strategies [20]. Particularly, carotenoids from microalgae exhibit a number of biological activities, including anticancer, antioxidants, antifungal, and antibacterial [20,21]. In recent years, in addition to the well-known carotenoids (e.g., astaxanthin, lutein, and β-carotene), synthesized by green microalgae, two other emerging carotenoids (i.e., fucoxanthin and peridinin) have gained attention due to their biological activities [22,23]. Peridinin is an apocarotenoid exclusively found in phototrophic dinoflagellates and contributes to light harvesting and protects cellular photosynthetic machinery from photo-oxidative damages by scavenging free radicals [23]. However, due to the difficulties in dinoflagellates isolation and cultivation, this carotenoid has not been widely studied in terms of biological activity.

The use of wastewater from shrimp rearing in synbiotic systems (AWW-SSs) was examined as growth culture medium for the dinoflagellate *Durusdinium glynnii*. First, the growth and nutrient uptake of *D. glynnii* were investigated and second, the resulting biomass was characterized in terms of (i) pigment and fatty acid composition and (ii) anti-*Vibrio* activity.

## 2. Materials and Methods

### 2.1. Shrimp Production

The wastewater from the cultivation of the Pacific white shrimp *Penaeus vannamei* in a synbiotic system (SS) was collected at the end of the production cycle. In short, *P. vannamei* was reared in a circular fiberglass tank (10 m^−3^) under a super intensive system without water exchange for 3 months. The production was performed in seawater (salinity of 35 PSU) that received inorganic fertilization with urea (4.5 g m^−3^ N), triple superphosphate (0.3 g m^−3^ P), and sodium silicate (0.23 g m^−3^ Si). After two days, organic fertilization was introduced by 12 applications of a commercial product for 24 h under anaerobic conditions, and subsequently maintained under aerobic conditions for the same time. The organic fertilizer was composed of rice bran (20 g m^−3^), molasses (2 g m^−3^), sodium bicarbonate (4 g m^−3^), and a microorganism-based product (0.05 g m^−3^), containing *Bacillus subtilis*, *B. licheniformis*, *Saccharomyces* sp., and *Pseudomonas* sp., at a total of 5.5 to 6.5 × 10^7^ CFU g^−1^. This synbiotic system had a final C:N ratio of 4.13. The shrimp production results are described elsewhere [24].

### 2.2. Dinoflagellate Strain and Culture Conditions

*Durusdinium glynnii* (clone BMK 211) was obtained from Culture Collection of Laboratório de Produção de Alimento Vivo (Recife, Brazil). The strain was maintained in seawater (30 PSU), previously filtered (0.45 μm), and sterilized (121 °C for 21 min), enriched with f/2 medium without Si. The cultures were kept in a room with a controlled temperature (22 ± 1 °C), under continuous irradiance (150 μmol photons m^−2^ s^−1^) and transferred to a newly fresh medium regularly.

The experimental cultures were grown under the above-described conditions in 250 mL Erlenmeyer flasks with a continuous irradiance of 300 μmol photons m^−2^ s^−1^. The cultures were aerated with sterile air at a rate of 0.05 vvm without CO_2_ addition in order to reduce production costs and to enable application on a large scale.

Aquaculture wastewater (AWW) was submitted to solids sedimentation for 30 min and the supernatant was double-filtered (1 μm pore size). Moreover, the water was chlorinated (50 ppm) and neutralized (25 ppm of thiosulphate solution). The seawater used to prepare f/2 medium was filtered, and the above-described purification, disinfection, and sterilization procedures were applied.

### 2.3. Experimental Design

Four proportions of aquaculture wastewater from synbiotic system (AWW–SS) were applied: 25 (25% AWW–SS), 50 (50% AWW–SS), 75 (75% AWW–SS), and 100% (100% AWW–SS), the remainder, i.e., 75, 50, 25, and 0%, respectively, were completed with f/2 culture medium (FM). In addition to these four treatments, the dinoflagellate was grown in culture medium without wastewater addition (positive control). All treatments and the control had three independent replicates, and the experiment was carried out in a completely randomized design.

### 2.4. Biological, Chemical, and Biochemical Analyses

Samples were taken for growth analyses at day 0, 1, 2, 3, 6, 9, 12, and 15 from each independent replicate. The samples taken at days 0, 7, and 15 were collected to determine nutrient uptake (i.e., nitrogen and phosphorus) and pigment content (chlorophylls and carotenoids) of *D. glynnii*.

#### 2.4.1. Growth Evaluation

Biomass (dry weight in g L^−1^) was estimated by the gravimetric method using 0.45 μm glass fiber microfilters. The maximum biomass reached (B_max_) and the yield (Y, mg L^−1^ day^−1^) were calculated [25]. Moreover, cell concentration (c, cells mL^−1^) was determined using a Neubauer chamber and optical microscopy. The specific growth rate (µ, day^−1^) in the exponential phase was calculated from cell concentrations as described in Oliveira et al. [25].

#### 2.4.2. Nitrogen and Phosphorus Analyses

Filtrated growth media (0.45 μm glass fiber microfilters) were analyzed for total N and P with the ammonium–N (N–NH_4_^+^; [26]), nitrite–N (N–NO_2_; [27]), nitrate–N (N–NO_3_; [28]), orthophosphate (P–PO_4_^3−^; [29]) methods. The removal efficiency (in %) was calculated according to Ansari et al. [27].

#### 2.4.3. Pigment Analysis

Aliquots of 15 mL were centrifugated at 3000 rpm for 10 min, and the remaining biomass was subjected to pigment extraction using acetone 90% [28]. Chlorophyll content (i.e., chlorophyll–*a* + *c*) was calculated according to Jeffrey and Humphrey [30] and carotenoid content (i.e., total carotenoids, β-carotene, and peridinin) was analyzed by following the methods proposed by Carreto and Catoggio [31] and Prézelin [32]. Values for all pigment concentrations were normalized to mg pigment (g biomass)^−1^.

#### 2.4.4. Lipid Extraction and Fatty Acid Composition

The total lipids were extracted following the Folch et al. (1957) method. In brief, a 2:1 chloroform–methanol (*v*/*v*) solution was added to approximately 0.2 g of sample and mixed in an Ultra-turrax^®^ for 2 min, then a 0.73% sodium chloride solution was added to achieve a 2:1:0.8 ratio of chloroform–methanol–water (*v*/*v*/*v*). Phase separation was achieved by centrifugation at 500× *g* for 5 min. The lipid phase was recovered and washed once with chloroform and further centrifuged.

The fatty acid compositions of the extracted lipids from *D. glynnii* biomass were determined after the conversion of fatty acids to their corresponding methyl esters using the method of O’Fallon et al. (2007) [33]. The analysis of fatty acid methyl esters (FAMEs) was performed on a gas chromatograph (model GC-2014, Shimadzu, Kyoto, Japan), equipped with split-injection-port flame-ionization detector, and 105 m long Restek capillary column (ID = 0.25 mm) coated with 0.25 μm of 10% cyanopropylphenyl and 90% biscyanopropylsiloxane. The chromatographic conditions and the method for identifying fatty acids were the same as those described in Oliveira et al. [11]. The FAMEs analyses were performed in duplicate (*n* = 2).

### 2.5. Antibacterial Activity

*Vibrio* strains (i.e., *V. parahaemolyticus* and *V. vulnificus*) were cultivated onto thiosulfate citrate bile salt agar (TCBS) plates and incubated at 37 °C. The antimicrobial susceptibility of the *V. parahaemolyticus* and *V. vulnificus* strains was determined using acetonic extract of *D. glynnii* biomass through the Kirby–Bauer method. A bacteria broth (10^8^ CFU mL^−1^) was seeded in the Petri plates (140 × 15 mm) containing the TCBS medium with the aid of a sterile swab. Sterile blank paper disks (6 mm diameter) impregnated with 20 µL of dried algae extracts (10%, *w v*^−1^) were added onto the agar plates, and subsequently incubated at 30 °C for 24 h. A disk with only solvent (i.e., 90% acetone) was used to determine the potential inhibition by the solvent. The inhibition zone of the solvent disk was subtracted from the inhibition of each of the extracts. A transparent ring around the paper disk revealed antibacterial activity, and transparent diameter was measured using a digital caliper to determine the inhibition zone.

### 2.6. Economic Analysis

An economic analysis was conducted assuming the cost differences amongst the culture media. Culture medium costs were calculated according to the final concentration of each element used for production of one kilogram of dry biomass. The prices of the f/2 medium used was based on Faé Neto [34]. The cost for the treatment of synbiotic wastewater was considered zero as the wastewater processing was like the seawater and no further addition of reagents were added to this medium.

### 2.7. Statistical Analysis

Single comparisons were performed using one-way ANOVA, followed by Tukey’s post hoc mean comparison test (normality of the data and homogeneity of the variances were previously verified, by the Shapiro–Wilk and Levene tests, respectively). In addition, linear and non-linear regressions were calculated to plot correlation between inhibition zone and peridinin content and growth curves of *D. glynnii* cultures subjected to different proportions of wastewater from shrimp culture, respectively. For all analyses, a level of significance of 5% was adopted.

## 3. Results

### 3.1. Growth Performance

In the present study, the endosymbiotic dinoflagellate *Durusdinium glynnii* was able to grow in pure AWW–SS, and partial AWW-SS replacement with FM above 25% improved its growth performance in comparison to the control (Figure 1A). The onset of the exponential growth phase (on the 5th day) was faster for the 50 and 75% AWW–SS proportions. The treatments 25% and 100% AWW–SS had a shorter exponential growth phase. All treatments, and the control, reached maximum cell density on the 12th day of cultivation. The 75% AWW–SS resulted in higher values of B_max_ (0.51 ± 0.04 g L^−1^) and Y (57.04 ± 4.63 mg L^−1^ day^−1^), in comparison to the other treatments and the control (0.33 ± 0.03 g L^−1^ and 37.00 ± 3.02 mg L^−1^ day^−1^) (Figure 1B). The partial replacement (i.e., 50 or 75% AWW–SS) of FM by AWW–SS, resulted in higher values of µ (0.56 ± 0.01 and 0.58 ± 0.01 day^−1^, respectively; Figure 1C).

### 3.2. Nutrient Removal

The different AWW–SS proportions evaluated in the present study resulted in different initial concentrations of N–NH_4_^+^, N–NO_2_^−^, N–NO_3_^−^, and P–PO_4_^3−^ (Figure 2). N–NO_3_^−^ was completely absorbed in the control and the 25%–AWW–SS, while for N–NO_2_^−^, low or no absorption was observed. In the 50% AWW–SS, the initial level of N–NH_4_^+^ was considerably higher than the final level. A high phosphorus uptake was observed in all treatments.

### 3.3. Pigments Composition

Overall, lower oscillation in pigment content was found at day 7 compared to day 15. At day 15, higher levels of chlorophyll were measured in cells grown on 50% AWW–SS compared to the 25% AWW–SS treatment and the control (Figure 3A). Similarly, high content of total carotenoids, β-carotene, and peridinin were found in cells grown at 50% AWW–SS at the end of the cultivation period (Figure 3B–D).

### 3.4. Fatty Acids Profile

The FAME profile of *D. glynnii* was considerably different amongst the applied treatments. The PUFA content ranged from 15.15% to 8.55% for the 100% AWW-SS and FM treatments, respectively. Notably, the DHA fatty acid content of the biomass cultivated in 100% AWW-SS reached 11.14% of the total fatty acids (equivalent to 2.76% of the dry weight when correlated with the lipid yield, which was 24.76% of the dry weight) (Table 1).

### 3.5. Antibacterial Activity

The acetonic extracts obtained from *D. glynnii* biomass introduced inhibitory effects on the *Vibrio parahaemolyticus* and *V. vulnificus* (Figure 4). An inhibition zone of about 10 mm for *V. parahaemolyticus* was measured for all the extracts regardless of the applied culture treatment. For *V. vulnificus*, an inhibition of about 6 mm was observed for the control, 25% and 50% AWW–SS, while a significative higher inhibition zone (close to 10 mm) was observed for the extracts from the biomass grown in 75% and 100% AWW–SS. Finally, the inhibition zone of *V*. *parahaemolyticus* was positively correlated (*p* < 0.05) with the peridinin content, while this correlation was not significant (*p* = 0.47) for *V. vulnificus*.

### 3.6. Economic Analysis

Based on the growth data, peridinin content, and culture medium cost, the economic analysis was conducted and summarized in Table 2. As the percentage of wastewater use increases, the cost of producing biomass and peridinin decreases. The treatment using only wastewater from shrimp culture was the cheapest, but due to its low biomass yield in comparison to the 75% AWW–SS treatment, it has a longer production time.

## 4. Discussion

The ability of microalgae to grow in aquaculture wastewater represents an important mechanism towards the development of aquaculture circular models. A number of microalgae species have already been effectively used to convert nitrogen and phosphorus from marine and freshwater aquaculture effluents into biomass, and some of these reports are listed in Table 3. However, most of these studies do not present a real usability of microalgal biomass, in order to promote circularity in aquaculture. Here, extracts from microalgal biomass produced with wastewater from a synbiotic system have demonstrated in vitro antibacterial activity against two pathogenic *Vibrio* bacteria strains. In addition to peridinin, other carotenoids (i.e., β-carotene, lycopene, and fucoxanthin) have been documented earlier as antibacterial agents [35,36]. Thus, it is possible that peridinin is primarily responsible for the anti-*Vibrio* effect, or that another carotenoid, or even the synergistic combination of various carotenoids, plays this role. Further studies are required to confirm these hypotheses.

In a comparative study of the marine microalgae *Chaetoceros muelleri*, *Nannochloropsis oculata*, and *Tetraselmis chuii* cultivated in shrimp biofloc wastewater, Magnotti et al. [37] reported that the biomass gains of *T. chuii* and *N. oculata* cultivated in the wastewater were similar to the gains from standard culture medium (f/2 medium). Although the synbiotic shrimp farming system is similar to the biofloc system, differences in inorganic and organic fertilization may explain in this study the dinoflagellate *D. glynnii* exhibited greater biomass gains in wastewater treatment compared to the f/2 medium. Moreover, an optimization study using response surface methodology demonstrated that pH, retention time, and initial algal density are important parameters for the growth of *Picochlorum maculatum* in wastewater from shrimp rearing [38].

The levels of nitrogen and phosphate compounds in aquaculture wastewater vary greatly between production systems, feed offered, animal density, etc. [27,37]. The biofloc-based system, i.e., heterotrophic, chemoautotrophic, and synbiotic system, accumulate large amounts of nitrate and orthophosphate over successive production cycles [42]. Although nitrate exhibits less toxicity to shrimp than other nitrogen forms, concentrations above 220 mg L^−1^ can reduce shrimp growth and survival [43]. Thus, nitrate decontamination is necessary to enable the reuse of water for several successive cycles. Various physical and chemical methods can be applied efficiently to transform nitrate to harmless nitrogen gas; however, they are expensive and do not add value to aquaculture systems [44,45]. Herein, we demonstrated a reduction of nitrate, and other inorganic compounds, concomitant with the valorization of the produced biomass during dinoflagellate induced bioremediation.

The removal efficiency of TP found in the present study was similar to those reported for marine diatoms and chlorophytes [18,27,46], but for TN the removal efficiency was lower. For example, a removal efficiency of TN of 86.1% (6.81 to 1.17 mg L^−1^ of N) from the Pacific white shrimp rearing by *Chlorella vulgaris*, using a membrane photobioreactor, was reported by Gao et al. [16]. Although some comparisons on removal efficiency may be subjective—that is, effluents with low initial concentrations of N and P tend to have higher removal rates—the physicochemical characteristics of the effluent after microalgae growth are considerably better and more suitable for disposal to aquatic ecosystems. In the present study, the levels of TN were reduced from 82.3 to 41.1 mg L^−1^, a nitrogen reduction five times higher than the reduction reported by Gao et al. [16].

Dinoflagellates are recognized for their high capacity to absorb and accumulate phosphorus, including organic and inorganic forms [46]. Thus, organic phosphorus available in synbiotic systems may have been remineralized by the bacterial community and was available to support the growth of *D. glynnii*. This fact may additionally support the higher growth of *D. glynnii* in wastewater-containing medium, compared to the control, since the enrichment of the phosphorus pool may contribute to the increase in the growth rate of dinoflagellates [47,48].

Partial replacement of the culture medium by 50 or 75% of AWW–SS improved the growth performance of *D. glynnii* compared to control and other treatments. In general, marine dinoflagellates show low growth rates due to shear stresses and other nutritional issues not yet elucidated [49,50]. For example, the addition of soil extract to dinoflagellate culture medium is a traditional method to improve the growth of these microalgae (e.g., [51,52]). Although the composition of the soil extract is almost never evaluated, it is likely that some of the oligoelements present in the soil extracts may also be present in the biofloc wastewater. Furthermore, endosymbiotic dinoflagellates feed on organic matter, such as bacteria and small microalgae. Jeong et al. [47] reported that free-living *Symbiodinium* spp. acquired more nitrogen from prey than the uptake of inorganic nitrogen from the f/2 medium. Thus, organic residues smaller than 1 µm (filtration used in the present study) may also have served as feed for *D. glynnii*.

The biodiversity of dinoflagellates undoubtedly represents an inexhaustible source of organic compounds; however, the chemodiversity of dinoflagellates remains poorly studied, partly due to the challenges associated with cultivating these organisms. Despite these challenges, an increasing number of new dinoflagellate secondary metabolites have been described in recent years, showing potential as antibacterial and anticancer agents. In addition to these metabolites, their high content of PUFAs, which are essential for various biological functions and have substantial health benefits, further highlights the importance of exploring these microorganisms. A high DHA content was reported in another study conducted by our group [11]. In comparison with other microalgae known for their high DHA content, the maximum DHA content achieved in *Isochrysis galbana*, under optimized conditions, was 15.03% of the total fatty acids. Thus, it is likely that non-toxic marine dinoflagellates could have the same potential for aquaculture nutrition as diatoms.

Vibriosis is a threatening bacterial disease that affects aquaculture production worldwide. Additionally, the presence of *Vibrio* spp. can cause gastrointestinal problems in humans [51]. Thus, new natural antibiotics are being investigated to contribute to the sustainable development of aquaculture, and some of these reports are summarized in Table 4. Soto-Rodriguez et al. [52] reported aqueous extract from the diatom *Chaetoceros calcitrans* inhibited the growth of *V*. *parahaemolyticus*. In the present study, the inhibition of *V*. *parahaemolyticus* was positively correlated with the content of the peridinin carotenoid. The mechanisms of antibacterial activity of carotenoids include cytoplasm leakage, nucleic acid formation inhibition, and outer membrane permeability. In addition, other carotenoids (e.g., β-carotene and fucoxanthin) have been documented earlier as antibacterial agents [35,36], thus, another carotenoid may have exhibited a greater influence on *V. vulnificus* inhibition than the peridinin one.

Here, we present a natural antibiotic (based on microalgae biomass and produced using residues from a synbiotic aquaculture system) that demonstrates the potential to control or mitigate vibriosis related diseases. In general, shrimp infected by *Vibrio* strains show a necrosis of the appendages, expanded chromatophores, an empty gut, and an absence of fecal strands [9,52], and the cumulative mortalities of infected shrimp may reach up to 80%, resulting in economic losses in the billions of dollars annually [52]. Therefore, a holistic and circular aquaculture model contributes to the solution of one of the main current threats to marine shrimp rearing.

## 5. Conclusions

The results of the present study clearly demonstrate that wastewater from a synbiotic system has adequate characteristics for the growth of the marine dinoflagellate *D. glynnii*, and it has shown higher biomass productivity when grown at a 75% AWW–SS. The nitrogen and phosphorus levels of wastewater were reduced by 50.1 and 71.7%, respectively, and this facilitated the reuse of the water for successive shrimp production cycles, reducing the negative impacts of harmful compound accumulation on the cultivated animals. However, it is important to note that these results may face diverse challenges when scaled up for field application, due to thermal variations, susceptibility to contamination, and other factors. Finally, metabolites from dinoflagellate biomass produced using synbiotic wastewater as a culture medium can help to control vibriosis during shrimp production. This circular approach represents a robust model towards the development of circularity in aquaculture, contributing to the achievement of the SDGs of the 2030 Agenda.

## Figures and Tables

**Figure 1 microorganisms-12-01598-f001:**
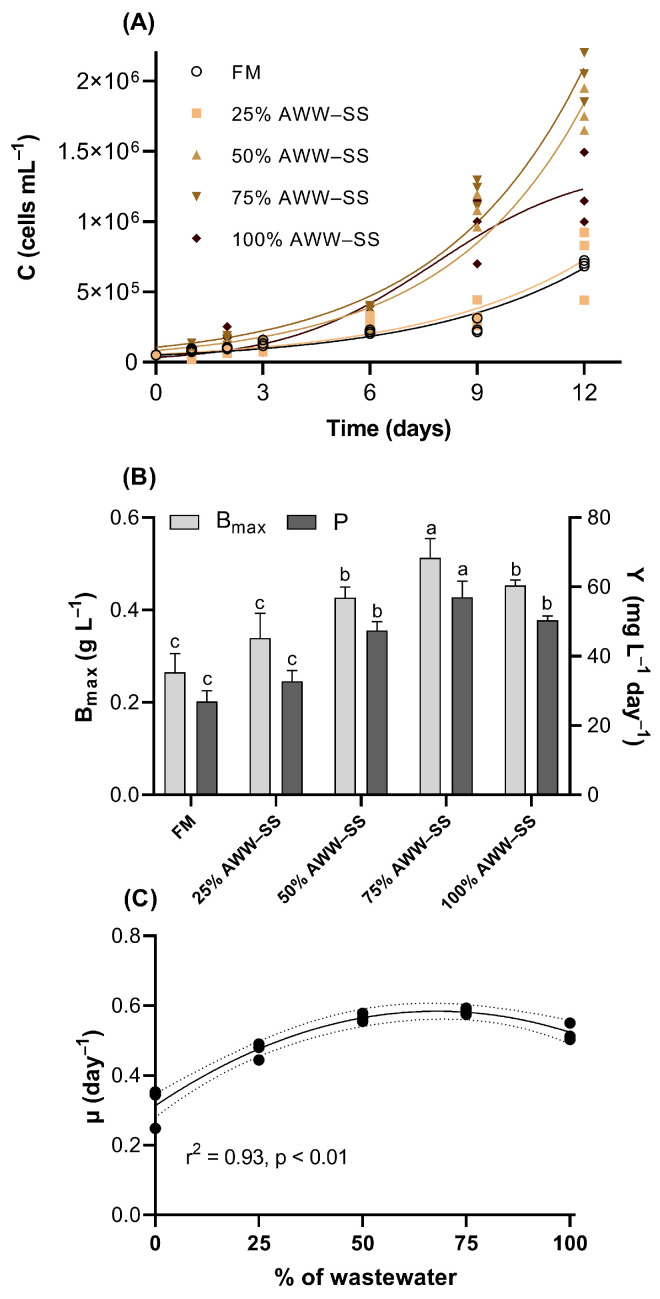
Logistic growth curves (**A**), maximum biomass reached and yield (**B**), and specific growth rates (**C**) of *Durusdinum glynnii* cultivated in different proportions of wastewater from shrimp culture in a synbiotic system. µ—specific growth rate (day^−1^), B_max_—maximum biomass reached (g L^−1^), Y—daily biomass productivity (mg L^−1^ day^−1^). Pointed lines represent 95% of confidence interval. Different letters indicate significant differences (*p* < 0.05) between treatments by Tukey’s post hoc test.

**Figure 2 microorganisms-12-01598-f002:**
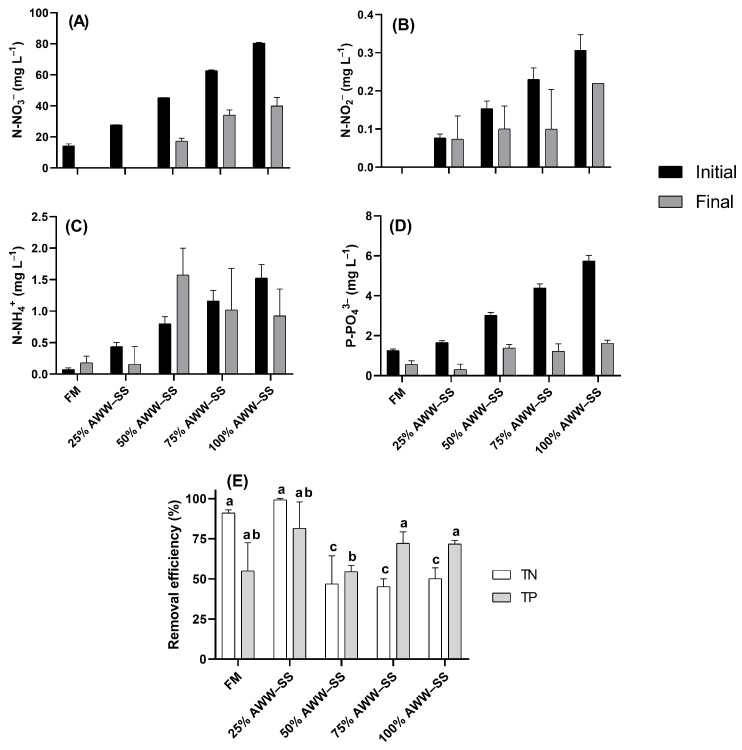
Nutrients uptake (**A**–**D**) and efficiency removal (**E**) by *Durusdinium glynnii* cultivated in different proportions of wastewater from shrimp culture in a synbiotic system. Different letters indicate significant differences (*p* < 0.05) between treatments by Tukey’s post hoc test.

**Figure 3 microorganisms-12-01598-f003:**
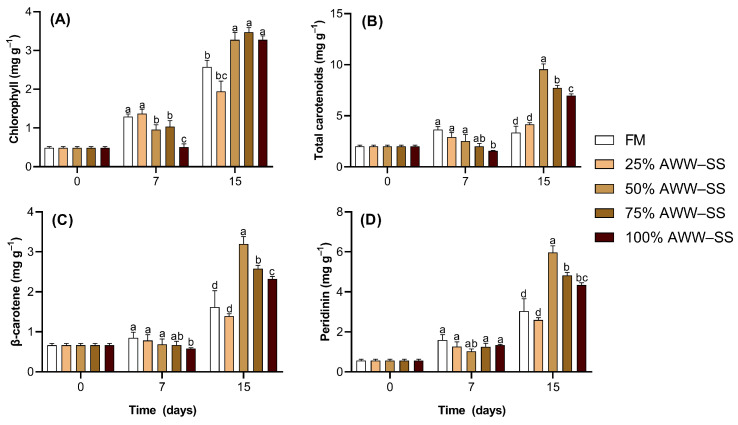
Chlorophyll (**A**), total carotenoids (**B**), β-carotene (**C**) and peridinin (**D**) contents of *Durusdinum glynnii* cultivated in different proportions of wastewater from shrimp culture in a synbiotic system. Different letters indicate significant differences (*p* < 0.05) between treatments by Tukey’s post hoc test.

**Figure 4 microorganisms-12-01598-f004:**
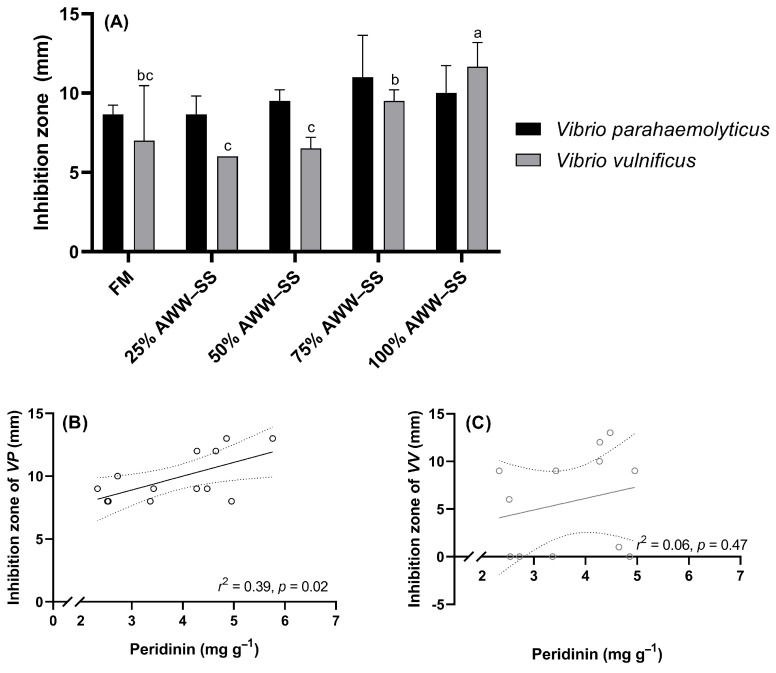
Inhibition zone (**A**) of extracts from *Durusdinium glynnii* biomass and regressions (**B**,**C**) with peridinin against *Vibrio parahaemolyticus* and *V. vulnificus* bacteria strains. *VP*—*Vibrio parahaemolyticus*; *VV*—*Vibrio vulnificus*. Pointed lines represent 95% of confidence interval. Different letters indicate significant differences (*p* < 0.05) between treatments by Tukey’s post hoc test.

**Table 1 microorganisms-12-01598-t001:** Fatty acid composition (%) of *Durusdinium glynnii* cultured in different proportions of wastewater from shrimp culture in a synbiotic system.

Fatty Acid	FM	25% AWW-SS	50% AWW-SS	75% AWW-SS	100% AWW-SS
C10:0	0.22 ± 0.10	0.56 ± 0.07	0.48 ± 0.10	0.70 ± 0.09	0.60 ± 0.12
C12:0	2.28 ± 0.23	4.50 ± 0.14	3.02 ± 0.19	4.48 ± 0.20	3.73 ± 0.22
C14:0	6.42 ± 0.32	9.73 ± 0.27	7.66 ± 0.44	12.10 ± 0.37	9.00 ± 0.57
C15:0	0.12 ± 0.05	0.18 ± 0.10	0.18 ± 0.10	0.21 ± 0.05	0.21 ± 0.10
C16:0	39.90 ± 0.47	34.16 ± 0.55	32.20 ± 0.25	33.46 ± 0.35	28.57 ± 0.65
C17:0	0.15 ± 0.03	0.17 ± 0.04	0.28 ± 0.05	0.34 ± 0.04	0.31 ± 0.07
C18:0	1.32 ± 0.13	1.28 ± 0.10	1.60 ± 0.15	1.75 ± 0.17	1.65 ± 0.12
C21:0	6.55 ± 0.22	8.92 ± 0.41	14.38 ± 0.37	11.52 ± 0.31	13.00 ± 0.18
Ʃ SFA	56.96 ± 0.78	59.50 ± 0.70	59.80 ± 0.69	64.56 ± 0.84	57.70 ± 077
C14:1	0.20 ± 0.05	0.15 ± 0.02	0.30 ± 0.04	0.23 ± 0.02	0.23 ± 0.01
C15:1	0.62 ± 0.11	0.51 ± 0.10	1.38 ± 0.12	1.24 ± 0.15	1.10 ± 0.10
C16:1	14.39 ± 0.11	9.36 ± 0.21	10.31 ± 0.12	10.18 ± 0.22	11.07 ± 0.18
C17:1	0.20 ± 0.05	0.15 ± 0.01	0.25 ± 0.02	0.26 ± 0.05	0.10 ± 0.04
C18:1	16.53 ± 0.47	15.03 ± 0.22	7.48 ± 0.14	6.58 ± 0.22	7.40 ± 0.17
C20:1	-	-	0.20 ± 0.03	0.14 ± 0.03	-
C22:1	-	0.11 ± 0.01	0.25 ± 0.03	0.20 ± 0.04	-
Ʃ MUFA	31.94 ± 0.41	25.20 ± 0.76	20.17 ± 0.29	18.83 ± 0.10	19.90 ± 0.22
C18:3 ω3 (ALA)	0.24 ± 0.05	0.43 ± 0.06	0.43 ± 0.08	0.53 ± 0.09	0.46 ± 0.05
C20:5 ω3 (EPA)	0.77 ± 0.03	0.90 ± 0.05	0.70 ± 0.03	0.81 ± 0.06	1.76 ± 0.06
C22:6 ω3 (DHA)	5.38 ± 0.21	7.70 ± 0.11	8.20 ± 0.18	6.81 ± 0.27	11.14 ± 0.16
Ʃ PUFA–ω3	6.40 ± 0.11	9.00 ± 0.09	9.33 ± 0.13	8.15 ± 0.16	13.36 ± 0.17
C18:2 ω6 (LA)	1.27 ± 0.05	1.16 ± 0.02	1.26 ± 0.02	1.00 ± 0.04	0.73 ± 0.02
C18:3 ω6 (GLA)	0.74 ± 0.02	0.18 ± 0.03	0.78 ± 0.07	0.46 ± 0.05	0.31 ± 0.03
C20:3 ω6	0.14 ± 0.05	0.16 ± 0.02	-	0.42 ± 0.03	0.30 ± 0.05
C20:4 ω6 (AA)	-	-	0.28 ± 0.02	0.21 ± 0.03	0.45 ± 0.04
Ʃ PUFA–ω6	2.15 ± 0.03	1.50 ± 0.02	2.32 ± 0.04	2.09 ± 0.03	1.79 ± 0.03
ω3/ω6	2.97	6.00	4.02	3.89	7.46

SFA = saturated fatty acid; MUFA = monounsaturated fatty acid; PUFA = polyunsaturated fatty acid.

**Table 2 microorganisms-12-01598-t002:** Economic analysis of *Durusdinium glynnii* production using different proportions of culture medium and wastewater from shrimp rearing in a synbiotic system.

Treatment	Culture Medium Cost (USD m^−3^)	Biomass Production (g m^−3^)	Peridinin Content (g kg^−1^)	USD per kg Biomass	USD per g Peridinin	Production Time (Days) *
FM	15.6	266.0 ± 39.7	3.0 ± 0.6	59.5 ± 8.7	20.2 ± 5.3	37.4 ± 4.1
25% AWW–SS	11.7	340.0 ± 52.9	2.6 ± 0.1	35.0 ± 5.9	13.5 ± 2.6	30.6 ± 2.7
50% AWW–SS	7.8	426.7 ± 23.1	6.0 ± 0.3	18.3 ± 1.0	3.1 ± 0.0	21.1 ± 1.2
75% AWW–SS	3.9	513.3 ± 41.6	4.8 ± 0.2	7.6 ± 0.6	1.6 ± 0.2	17.6 ± 1.4
100% AWW–SS	0	453.3 ± 11.5	4.3 ± 0.1	0	0	19.9 ± 0.5

* Time for production of 1 kg of biomass using a system with useful volume of 1 m^3^.

**Table 3 microorganisms-12-01598-t003:** Main characteristics of microalgae cultivation in various types of aquaculture wastewater.

Microalgae	Systems	Target Species	TN (%)	TP (%)	Refs.
*Durusdinium glynnii*	SS	Pacific white shrimp	50.1	71.7	This study
*Chaetoceros muelleri*	BFT	Pacific white shrimp	-	100	[37]
*Chlamydomonas* sp.	-	Tilapia	79.6	96.0	[38]
*Chlorella minutissima*	RAS	Salmon	88.0	99.0	[39]
*Chlorella vulgaris*	BFT	Tilapia	84.3	48.3	[18]
	RAS	Tilapia	99.8	82.7	[16]
	-	Pacific white shrimp	86.1	82.7	[16]
	-	Flathead grey mullet	95.4	92.0	[40]
*Isochrysis galbana*	-	Flathead grey mullet	66.0	91.9	[40]
*Nannochloropsis oculata*	BFT	Pacific white shrimp	83.0	100	[37]
*Picochlorum maculatum*	-	Pacific white shrimp	66.7	92.8	[41]
*Platymonas subcordiformi*	-	Flounder	100	100	[12]
*Spirulina* sp.	-	Tilapia	81.1	100	[17]
*Tetradesmus obliquus*	RAS	Tilapia	99.7	99.6	[13]
	RAS	Tilapia	80.1	~100	[27]
*Tetraselmis chuii*	BFT	Pacific white shrimp	87.0	100	[37]

BFT—biofloc system; RAS—recirculating aquaculture system; SS—synbiotic system; TN—total nitrogen; TP—total phosphorus.

**Table 4 microorganisms-12-01598-t004:** In vitro activity of some biological anti-*Vibrio* spp. agents.

Source	Type of Inclusion	Dosage (μg mL^−1^)	Method	*Vibrio* Strain	Refs.
Microalgae					
*Durusdinium glynnii*	AcE		KBM	*VP*, *VV*	This study
*Chaetoceros calcitrans*	AqE	70	LM	*VP*	[52]
Seaweeds					
*Caulerpa sertularioides*	ME	1000	MM	*VA*, *VP*	[53]
*Gracilaria fisheri*	CPE	50	LM	*VP*	[54]
*Gracilaria verrucosa*	EE	2	AD	*VH*	[55]
*Ulva lactuca*	ME	>1500	MM	*VA*, *VP*	[53]
Plants					
*Moringa oleifera*	EE	64	BMPA	*VA*	[56]
*Musa acuminata*	AqE	1560	DD	*VP*, *VA*	[57]
*Ocimum basilicum*	AqE	19	KBM	*VH*, *VP*, *VA*	[58]

AD—agar disk; DD—disk diffusion; KBM—Kirby–Bauer method; BMPA—broth microtiter plate assay; LM—liquid medium; MM—microplate methods; AcE—acetonic extract; AqE—aqueous extract; CPE—crude protein extract; EE—ethanolic extract; ME—methanolic extract; *VP*—*Vibrio parahaemolyticus*; *VH—Vibrio harveyi; VA—Vibio alginolyticus*; *VV—Vibrio vulnificus*.

## Data Availability

The data presented in this study are available on request from the corresponding author due to the fact that most of the data was processed using different software and contains information in Portuguese (the authors’ native language). This change is requested to avoid confusion in the interpretation of the data.

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
