# Peer review of "A Holistic Approach to Producing Anti-Vibrio Metabolites by an Endosymbiotic Dinoflagellate Using Wastewater from Shrimp Rearing"

_microorganisms, 2024, doi:10.3390/microorganisms12081598_

Round 1

Reviewer 1 Report (Previous Reviewer 2)

Comments and Suggestions for Authors

I agree with the first version of the work. I consider the information about the change in lipid and fatty acid content to be superfluous. However, if this is a requirement for publication, it is acceptable. I find the research results informative with great potential for application. I hope the research will be continued. 

Author Response

# Reviewer 1

  1. I agree with the first version of the work. I consider the information about the change in lipid and fatty acid content to be superfluous. However, if this is a requirement for publication, it is acceptable. I find the research results informative with great potential for application. I hope the research will be continued.

Authors response: The authors appreciate the positive feedback on the manuscript. Regarding the results of the fatty acid profile, the authors chose to include it due to the lack of biochemical information on dinoflagellates cultured under different conditions. This information can be useful towards the use of dinoflagellate biomass as an aquaculture input.

Reviewer 2 Report (Previous Reviewer 3)

Comments and Suggestions for Authors

In their response , the authors agreed with most critical comments. I don't change the opinion and cannot support publishing although some technical corrections were made. The antimicrobial activity has too poorly been proved, and this problem cannot be solved over a short time since further experiments are required. Adding fatty acid profile to the revised ms is artificial and is not supportive for the central idea.

Comments on the Quality of English Language

I am under impression that the text was not completely corrected, especially in microbiology part.

Author Response

# Reviewer 2

Comment 1: In their response, the authors agreed with most critical comments. I don't change the opinion and cannot support publishing although some technical corrections were made. The antimicrobial activity has too poorly been proved, and this problem cannot be solved over a short time since further experiments are required. Adding fatty acid profile to the revised ms is artificial and is not supportive for the central idea.

Authors response: The authors understand the reviewer's opinion and agree that there would not be enough time to redo the entire antibacterial analysis (and particularly impossible as we no longer have the biomass and/or extracts). Regarding the fatty acid profile, although it is not the central theme of the manuscript, the data can help highlight that dinoflagellate biomass can also be useful from a nutritional standpoint in aquaculture, contributing to the sector's sustainability. Finally, the authors appreciate the suggestions, which will be considered in future experiments.

Comments on the Quality of English Language: I am under impression that the text was not completely corrected, especially in microbiology part.

Authors response: As for the grammar, the authors have made an effort to carefully review the manuscript and rewrite sentences that might have been confusing.

Reviewer 3 Report (New Reviewer)

Comments and Suggestions for Authors

This is a very interesting paper that focuses on the aquaculture industry and some environmental issues that should be considered. The algae Durusdinium glynnii was cultured and its growth was evaluated in aquaculture wastewater from shrimp farming diluted in different proportions of growth medium. Subsequently, the removal of total nitrogen and total phosphorus was investigated. In addition, the biomass of Durusdinium glynnii was characterized with regard to its pigment and fatty acid composition as well as its anti-Vibrio activity. It was found that biomass extracts of the algae inhibit the growth of Vibrio parahaemolyticus and Vibrio vulnificus.

All sections of the paper are well written. The introduction and aim of the research are clear, the materials and methods are well explained, the results are well presented, as is the discussion, where it is particularly beneficial to make comparisons with other studies, that are listed in Table 3 and Table 4.

Comments on the Quality of English Language

English language is fine, only minor editing of English language is required

Author Response

# Reviewer 3

Comment 1: This is a very interesting paper that focuses on the aquaculture industry and some environmental issues that should be considered. The algae Durusdinium glynnii was cultured and its growth was evaluated in aquaculture wastewater from shrimp farming diluted in different proportions of growth medium. Subsequently, the removal of total nitrogen and total phosphorus was investigated. In addition, the biomass of Durusdinium glynnii was characterized with regard to its pigment and fatty acid composition as well as its anti-Vibrio activity. It was found that biomass extracts of the algae inhibit the growth of Vibrio parahaemolyticus and Vibrio vulnificus.

All sections of the paper are well written. The introduction and aim of the research are clear, the materials and methods are well explained, the results are well presented, as is the discussion, where it is particularly beneficial to make comparisons with other studies, that are listed in Table 3 and Table 4.

Authors response: The authors appreciate the positive feedback on the manuscript.

Comments on the Quality of English Language: English language is fine, only minor editing of English language is required

Authors response: The English grammar was carefully review in the entire manuscript and we have rewritten sentences that might have been confusing.

Round 2

Reviewer 2 Report (Previous Reviewer 3)

Comments and Suggestions for Authors

No further comments

Comments on the Quality of English Language

Please check minor mistakes, for example: media... was (it is not correct).

This manuscript is a resubmission of an earlier submission. The following is a list of the peer review reports and author responses from that submission.

Round 1

Reviewer 1 Report

Comments and Suggestions for Authors

A Holistic Approach To Produce Anti–Vibrio Metabolites by an Endosymbiotic Dinoflagellate Using Wastewater From Shrimp Farming is promising. However, various grammatical problems are in the manuscript.

The hypothesis should be more clear.

All the figure should be given as significance level with star (*) marks.

Antibacterial activity

Discussion needs to be improved.

This form of manuscript cannot be published prior to revision. Therefore, I recommended it for major revision.

Comments on the Quality of English Language

Various grammatical problems are in the manuscript.

Reviewer 2 Report

Comments and Suggestions for Authors

The paper analyzes a study proposing an innovative and holistic method to address environmental challenges in aquaculture, using the endosymbiotic dinoflagellate Durusdinium glynnii to treat wastewater from shrimp farming. This approach aligns with sustainable development goals, promoting the principles of a circular bioeconomy.

The experiments were comprehensively planned, analyzing multiple aspects of the results obtained.

Several questions have arisen, and addressing them could improve the study's content:

1. Did you consider the variability in the composition of wastewater resulting from shrimp farming? Were the waters used in the experiments collected at the end of an industrial cycle? If so, this should be mentioned.

2. Why did you use the optimized f/2 medium as the control for growing D. glynnii instead of natural seawater, which might have been more appropriate? If the reason was to maintain mixotrophic conditions, this should be noted in the text.

 3. There is a question regarding the light intensity used: 150 μmol photons/m²/s for maintaining the microalgal culture and 300 μmol photons/m²/s for the experimental variant. Can you explain this manipulation?

4. It appears that specific conditions were established to synthesize pigments, particularly peridinin. Consequently, the objective should highlight both the purification process and the technological aspects of peridinin production. This is especially relevant because the economic analysis includes factors aimed at reducing the cost of biomass and peridinin production. Respectively, the conclusions will be reformulated.

5. Although the study reports improved growth rates and biomass yields in certain proportions of wastewater, the authors do not compare these results with other microalgae or dinoflagellates under similar conditions. A comparative analysis could provide a better understanding of the relative performance of D. glynnii.

6. From the perspective of wastewater purification, the study is highly relevant. It shows promising results under controlled laboratory conditions, but potential challenges for technological processes in commercial applications are not discussed. It would be beneficial to add a discussion about the types of purification systems using microalgae as a component of shrimp farms proposed as a result of this study.

It would be beneficial for the authors to mention all the questions that arose during the study, such as optimizing conditions for efficient nitrogen removal.

Addressing these questions could provide clarity and potentially enhance the overall quality and impact of the study.

Reviewer 3 Report

Comments and Suggestions for Authors

In their ms, the authors have report the differences in the growth parameters and nitrogen and phosphorus removal for dinoflagellate Durusdinium glynnii depending upon the cultivation conditions: in the commonly applied medium or aquaculture wastewater with so-called symbiotic system. Unfortunately, this ms suffers from serious flaws and cannot be recommended for publication in Microorganisms.

 Major Criticism

 Overall, the microbiological methods are outdated and too simple.

 The major weakness is that the antimicrobial activity of crude extracts has yet been tested too preliminarily to claim anti-Vibrio effect.

The composition of constituents is still unknown, and it is not ruled out that other metabolites to be extracted with acetone are active, not carotenoids only. Furthermore, the antimicrobial action of pigments if any has not been confirmed in this study: samples of individual pigments should be assayed too.

Differences in inhibition zones for the control (i.e., solvent) and extracts are not convincing.

A positive correlation of inhibition zones with an elevated content of piredinin is not enough.

It is necessary to test as the additional control the extract from growth medium.

Biological and chemical methods are insufficiently described and must be given as much complete as possible.

Economic analysis is redundant for a paper in microbiology-oriented journal.

Discussion is not well balanced with the results of this study.

Surely, the authors should consult the expert in microbiology to improve and correct the text. There are mistakes in terminology. Some of them are listed below.

Why is the studied dinoflagellate termed as endosymbiotic?

What is “synbiotic system”?  The precise definition is necessary to include.

The authors have not described why this SS was used.

 Other comments

Biomass cannot inhibit (see Abstract)/

It is incorrect to use “culture  (of microorganisms) is conducted. Another verb construction must be chosen.

It is rough to write “seawater was submitted”.

Unclear: “fertilization started through 12 applications of product for 24 h in an anaerobic phase followed by an aerobic phase (24 h).

What is “ a microbiome-based product containing Bacillus subtilis, B. licheniformis… Whose microbiome?

Neubaurer chamber is the appropriate name instead hemocytometer

A bacteria broth cannot form a solid medium in Petri dishes.

Unclear:  Sterile blank paper discs (6 mm diameter) impregnated with 20 μL of extracts carried out  (???) using dried algae (10%, w v−1) were added onto agar plates.

What are aliquots from biomass filtering?

Not Bmax, the term Ymax is commonly used.

English in not appropriate and should be improved substantially.

Comments on the Quality of English Language

The experimental section is too poorly written with numerous mistakes. I found some of them in Comment to Authors.